# Diagnostic Performance of On-Site Computed Tomography Derived Fractional Flow Reserve on Non-Culprit Coronary Lesions in Patients with Acute Coronary Syndrome

**DOI:** 10.3390/life12111820

**Published:** 2022-11-08

**Authors:** Abdelkrim Ahres, Judit Simon, Balazs Jablonkai, Bela Nagybaczoni, Tamas Baranyai, Astrid Apor, Marton Kolossvary, Bela Merkely, Pal Maurovich-Horvat, Balint Szilveszter, Peter Andrassy

**Affiliations:** 1Department of Cardiology, Bajcsy-Zsilinszky Hospital, Maglodi Rd. 89-91., H-1106 Budapest, Hungary; 2MTA-SE Cardiovascular Imaging Research Group, Heart and Vascular Center, Semmelweis University, Varosmajor Str. 68., H-1222 Budapest, Hungary; 3Medical Imaging Center, Semmelweis University, Ulloi Rd. 78a., H-1082 Budapest, Hungary

**Keywords:** acute coronary syndrome, non-culprit lesions, on-site computed tomography derived fractional flow reserve, invasive fractional flow reserve, dobutamine stress echocardiography

## Abstract

The role of coronary computed tomography angiography (CCTA) derived fractional flow reserve (CT-FFR) in the assessment of non-culprit lesions (NCL) in patients with acute coronary syndrome (ACS) is debated. In this prospective clinical study, a total of 68 ACS patients with 89 moderate (30–70% diameter stenosis) NCLs were enrolled to evaluate the diagnostic accuracy of on-site CT-FFR compared to invasive fractional flow reserve (FFRi) and dobutamine stress echocardiography (DSE) as reference standards. CT-FFR and FFRi values ≤0.80, as well as new or worsening wall motion abnormality in ≥2 contiguous segments on the supplying area of an NCL on DSE, were considered positive for ischemia. Sensitivity, specificity, positive, and negative predictive value of CT-FFR relative to FFRi and DSE were 51%, 89%, 75%, and 74% and 37%, 77%, 42%, and 74%, respectively. CT-FFR value (β = 0.334, *p* < 0.001) and CT-FFR drop from proximal to distal measuring point [(CT-FFR drop), β = −0.289, *p* = 0.002)] were independent predictors of FFRi value in multivariate linear regression analysis. Based on comparing their receiver operating characteristics area under the curve (AUC) values, CT-FFR value and CT-FFR drop provided better discriminatory power than CCTA-based minimal lumen diameter stenosis to distinguish between an NCL with positive and negative FFRi [0.77 (95% Confidence Intervals, CI: 0.67–0.86) and 0.77 (CI: 0.67–0.86) vs. 0.63 (CI: 0.52–0.73), *p* = 0.029 and *p* = 0.043, respectively]. Neither CT-FFR value nor CT-FFR drop was predictive of regional wall motion score index at peak stress (β = −0.440, *p* = 0.441 and β = 0.403, *p* = 0.494) or was able to confirm ischemia on the territory of an NCL revealed by DSE (AUC = 0.54, CI: 0.43–0.64 and AUC = 0.55, CI: 0.44–0.65, respectively). In conclusion, on-site CT-FFR is superior to conventional CCTA-based anatomical analysis in the assessment of moderate NCLs; however, its diagnostic capacity is not sufficient to make it a gatekeeper to invasive functional evaluation. Moreover, based on its comparison with DSE, CT-FFR might not yield any information on the microvascular dysfunction in the territory of an NCL.

## 1. Introduction

Patients with acute coronary syndrome (ACS) often present with angiographically moderate non-culprit lesions (NCL) [1]. Several clinical trials verified the superiority of complete revascularization over culprit-only intervention strategy in ACS patients with multivessel coronary artery disease [2,3,4]. However, the adequate method and the timing to determine the need of revascularization of an NCL remains controversial [5,6,7,8]. In recent times, coronary computed tomography angiography (CCTA) has become a useful non-invasive tool to exclude significant coronary artery disease in patients with non-ST-elevation acute coronary syndrome (NSTE-ACS) at low-to-intermediate risk according to its high negative predictive value (NPV) [5,9]. Considering that CCTA has a poor positive predictive value (PPV) [10] in the identification of an ischemia causing coronary stenosis and the decision for further treatment strategy is based only on anatomical imaging, a functional supplementation of CCTA has become needed. Non-invasive off-site computation of fractional flow reserve (FFR) from CCTA was reported to have excellent diagnostic performance compared to invasively measured FFR (FFRi) for the identification of lesion-specific ischemia in patients with stable angina [11,12,13]. Recently, on-site CCTA derived FFR (CT-FFR) simulation tools have been applied in contemporary stable angina patients, yielding better diagnostic capacity than luminal stenosis assessment with excellent inter- and intra-reader reproducibility [14]. However, limited data exists regarding both methods as to their clinical utility in ACS patients with concomitant moderate NCLs [15]. In contrast, FFRi and non-invasive dobutamine stress echocardiography (DSE) are widely validated and recommended methods for the functional interpretation of NCLs in ACS patients [16,17,18,19,20]. In this investigator-initiated prospective clinical study, we aim to evaluate the diagnostic performance and limitations of on-site CT-FFR to identify lesion-specific ischemia compared to FFRi and DSE in patients with recent ACS accompanied by at least one angiographically moderate NCL. Furthermore, the secondary aim was to investigate the potential superiority of CT-FFR over conventional CCTA-based anatomical assessment in this clinical setting.

## 2. Materials and Methods

### 2.1. Patient Selection

Patients between March 2017 and March 2019 with ACS and ≥1 moderate NCL [defined as 30–70% minimal lumen diameter stenosis (MLDS%) by invasive quantitative coronary angiography (QCA) [21] at the event of ACS] suitable for revascularization were prospectively enrolled. Screening was started at the time of the primary percutaneous coronary intervention (PCI); enrollment was done before discharge of the patients. ACS [(ST-elevation myocardial infarction (STEMI) and NSTE-ACS] was determined in accordance with the recommendations of the European Society of Cardiology (ESC) [5,8,22]. Exclusion criteria were: 1. age <18 years and >80 years, 2. left main coronary artery stenosis (≥50% by QCA), 3. severe valvular disease that needs surgical or percutaneous intervention, 4. co-dominant coronary circulation, 5. coronary anomalies, 6. ACS with non-obstructive coronary artery disease, 7. MLDS% of NCL >70% or <30% by QCA, 8. chronic total occlusion of the NCL, 9. history of severe and/or anaphylactic contrast reaction during any CT-scan/invasive angiography with unknown contrast agent, 10. chronic kidney disease (stage 4 or worse), 11. inability to cooperate with CT-scan acquisition and/or breath hold instructions, 12. insufficient image quality on DSE despite using contrast agent, 13. insufficient image quality on CT-scan for CT-FFR simulations, 14. inability to provide informed consent, 15. known non-cardiovascular comorbidity with poor prognosis (life expectancy <2 years), 16. pregnancy, 17. known genetically determined cardiomyopathy, and 18. any type of adult congenital heart disease. At least 30 days after the index episode [15], all patients underwent conventional CCTA scan supplemented by on-site CT-FFR simulations and a DSE prior to a staged coronary angiography complemented by FFRi measurements. All tests were performed within a 30-day interval relative to each other. Imaging cardiologists were blinded to the invasive coronary angiography (ICA) results and interventionalists were blinded to the non-invasive tests.

### 2.2. Coronary Computed Tomography Angiography

All CCTA examinations were performed with a 256-slice scanner (Brilliance iCT, Philips Healthcare, Best, The Netherlands) with prospective ECG-triggered axial acquisition mode. For CCTA, 100–120 kV with 200–300 mAs tube current was used based on body mass index. Image acquisition was performed with 128 × 0.625 mm detector collimation and 270 msec gantry rotation time. For heart rate control, a maximum of 50–100 mg metoprolol was given orally or 5–20 mg intravenously, if necessary. In patients with a heart rate of < 80/min, end-diastolic triggering was applied with 3–5% padding (73–83% of the R-R interval), and in those with ≥ 80/min, end-systolic triggering was chosen (35–45% of the R-R interval). Iomeprol contrast material (Iomeron 400, Bracco, Milan, Italy) was used with 85–95 mL contrast agent at a flow rate of 4.5–5.5 mL/s from antecubital vein access via 18-gauge catheter using a four-phasic protocol. Bolus tracking in the left atrium was used to obtain proper scan timing. An amount of 0.8 mg sublingual nitroglycerin was given prior to CTA, unless systolic blood pressure was less than 100 Hgmm. CCTA data sets were reconstructed with 0.8 mm slice thickness and 0.4 mm increment.

### 2.3. CCTA Segmentation and CT-FFR Simulation

Coronary centerlines were first extracted automatically using an advanced cardiac application (Comprehensive Cardiac Analysis and CT-FFR, IntelliSpace Portal Version 9.0.1, Philips Healthcare, Cleveland, OH, USA). Once the centerlines were extracted, lumen segmentation was performed. When it was necessary, centerlines and segmentation were manually corrected by a single reader, based on our prior inter- and intra-reader observer agreement data [14]. Once the lumen segmentation was reviewed and corrected, qualitative and categorical stenosis was assessed per vessel, as recommended by the Society of Cardiovascular Computed Tomography [23]. Lesion length was measured as the distance between the first and last cross-sections of a vessel segment where the investigated plaque was unequivocally identifiable. The luminal diameter stenosis was quantified using a dedicated software tool by identifying the minimum diameter and the reference diameter for all stenosis (Comprehensive Cardiac Analysis, IntelliSpace Portal, Philips Healthcare, Cleveland, OH, USA). The mean diameter percentage is provided by the software, which corresponds to the minimum lumen diameter normalized to the reference points [24]. The segmented coronary artery tree lumen was used as an input to a research prototype CT-FFR simulation algorithm (Version 1.0.2, Philips Healthcare, Cleveland, OH, USA). The on-site CT-FFR application using a lumped parameter model allows for robust simulations integrating patient-specific boundary conditions. The manually adjusted lumen segmentation is entered to the lumped parameter simulator, where the vessel tree is converted into a network of equations to simulate pressure drops along the coronary vasculature. In the current protocol, correspondingly to the prior study where we tested the diagnostic accuracy of on-site CT-FFR estimation in stable angina patients as compared with FFRi [14], proximal and distal end of plaques were marked and CT-FFR values were measured distally from the lesion similarly to invasive measurements. CT-FFR value ≤0.80 was considered ischemic [13]. CT-FFR drop from proximal to distal measuring point [(CT-FFR drop), measured as CT-FFR value proximal minus CT-FFR value distal)] was also determined [25]. Figure 1 shows a representative example of on-site CT-FFR simulation from the present study.

### 2.4. Invasive Fractional Flow Reserve

During the acute ICA, primary PCI and anatomical screening (QCA) for eligible NCLs was performed according to the international standards [5,8,26]. FFRi measurement of NCLs was performed only as a part of the staged ICA procedure by using intracoronary nitrate and intravenous adenosine (140–180 µg/kg/min) administration after advancing the pressure wire (Pressure Wire X, St. Jude Medical, Saint Paul, USA or Comet, Boston Scientific, Marlborough, MA, USA) into the distal portion of the evaluated vessel [27]. FFRi ≤ 0.80 was considered functionally relevant [16,28].

### 2.5. Dobutamine Stress Echocardiography

DSE procedures were performed, and procedural endpoints were determined in line with the ESC expert consensus document [29]. At least 24 h before the examination, negative chronotropic medications were omitted. The 16-segment left ventricle model was used with intravenous contrast enhancement (Sonovue, Bracco, Italy) in patients with poor image quality. Coronary supplying areas and myocardial segments were specified in accordance with the ESC recommendation [30]. Categorical echocardiographic finding of ischemia on the supplying area of an NCL was defined as new or worsening wall motion abnormality (WMA) in ≥2 contiguous myocardium segments (ischemia at a distance) [19,29]. Furthermore, regional wall motion score index (WMSI) at peak stress on the non-culprit area, which is superior to global measurements in ischemia detection, was also calculated from segmental scores in agreement with the literature [31,32]. DSE images were analyzed by two blinded expert echocardiographers. A comprehensive description of DSE protocol can be seen in the Appendix A.

### 2.6. Statistical Analysis

Categorical variables were described as numbers and percentages; continuous variables were expressed as mean ± standard deviation. Normality of distribution was investigated by the Kolmogorov–Smirnov test. Continuous variables were compared with paired Student t-test or Mann–Whitney U test, as well as with one-way ANOVA or Kruskal–Wallis test (in cases of 3 groups), as appropriate. Chi-square test was applied to analyze the accuracy, sensitivity, specificity, PPV, and NPV of dichotomous CT-FFR against binary FFRi and DSE. Depending on the distribution, correlation analyses between continuous variables were made using Pearson or Spearman’s correlation coefficients. In order to determine the CCTA-based anatomical and functional predictors of FFRi and regional WMSI at peak stress, univariate and multivariate linear regression analysis was performed. Statistical significance for each parameter to be entered into the multivariate regression model was *p* < 0.1. Since CT-FFR value and CT-FFR drop are strongly correlated parameters (Appendix A), we planned to include them separately into different multivariate models to avoid multicollinearity. Furthermore, areas under receiver operating characteristic (ROC) curves (AUC) were calculated to evaluate the diagnostic ability of conventional CCTA-based anatomical stenosis markers, the different CT-FFR functional parameters, and their combined models to distinguish between functionally relevant and non-relevant NCL(s) defined by the reference methods. AUC values were compared using DeLong’s test [33]. A *p* < 0.05 was considered of statistical significance. SPSS Statistics 25 (IBM, Armonk, NY, USA), MedCalc v.20.011 (MedCalc Software, Ostend, Belgium) and GraphPad Prism 8 (GraphPad Software, San Diego, CA, USA) were used for statistical analysis.

## 3. Results

### 3.1. Study Population

During the enrollment period, a total of 68 patients (45 STEMI and 23 NSTE-ACS) with 89 NCLs [57 in patients with STEMI and 32 in patients with NSTE-ACS, lesion distribution was 52 LAD (left anterior descending coronary artery, 15 LCx (left circumflex coronary artery), and 22 RCA (right coronary artery), respectively] met the inclusion criteria (Figure 2). Due to anatomical reasons (small, heavily calcified vessel), primary PCI of the culprit lesion was not performed in one case (1.5%). Twenty enrolled patients (29.4%) had more than one eligible NCL. In this cohort, global anatomical complexity of coronary lesions was generally low (SYNTAX score before primary PCI: 18.5 ± 7.0, residual SYNTAX score after primary PCI: 10.2 ± 5.9). Baseline clinical data are presented in Table 1.

### 3.2. FFRi and DSE

In cases of 35 NCLs (39.3%), FFRi value was ≤0.80. MLDS% of NCLs measured by QCA decreased significantly from the initial ICA to the time of the staged invasive procedure (51.0% ± 8.1% vs. 44.9% ± 8.2%, *p* < 0.001). During the DSE procedures, ischemia was provoked in 27 NCL-supplied myocardium areas (30.3%) without any serious adverse event. The time [days, median (interquartile range, IQR)] between ACS and DSE, ACS and FFRi, as well as DSE and FFRi was 67.5 days (IQR: 51.25–86.75 days), 84 days (IQR: 71–103 days), and 13 days (IQR: 7–25 days), respectively. Detailed FFRi and DSE results are presented in Appendix A.

### 3.3. CT-FFR versus FFRi

All patients had good or excellent image quality with adequate heart rate during the CCTA-scans. Major anatomical and functional CCTA data (overall and divided into subgroups according to major epicardial arteries) are described in Table 2. MLDS% of NCLs by CCTA was lower than those measured by QCA during the staged ICA procedure (39.4% ± 16.2% vs. 44.9% ± 8.2%, *p* = 0.004). CT-FFR values were higher than FFRi values (0.85 ± 0.09 vs. 0.83 ± 0.08; *p* = 0.003). Twenty-four NCLs (27%) were determined functionally significant during CT-FFR simulations. The accuracy, sensitivity, specificity, PPV, and NPV of on-site CT-FFR to detect an NCL with ischemic FFRi were 74%, 51%, 89%, 75%, and 74%, respectively (Appendix A). Sensitivity and NPV was better in patients with STEMI than in patients with NSTE-ACS (60% vs. 40% and 80% vs. 63%, respectively) (Appendix A). Although lesion length did not correlate with FFRi value (r = −0.09, *p* = 0.402), MLDS%, CT-FFR value, and CT-FFR drop showed a modest correlation with it (r = −0.26, *p* = 0.015, r = 0.54, *p* < 0.001 and r = −0.54, *p* < 0.001, respectively). Both CT-FFR value (Model 1) and CT-FFR drop (Model 2)—included separately into the models due to multicollinearity issues—were predictive for FFRi value in multivariate analysis (β = 0.334, *p* < 0.001 and β = −0.289, *p* = 0.002, respectively). Detailed description is presented in Table 3 and Appendix A. As a result of comparing their AUC values, the discriminative power of CT-FFR value and CT-FFR drop was better than that of CCTA-based MLDS% in identifying an NCL with an FFRi ≤ 0.80 [0.77 (95% Confidence Intervals, CI: 0.67–0.86) and 0.77 (CI: 0.67–0.86) vs. 0.63 (CI: 0.52–0.73), *p* = 0.029, and *p* = 0.043, respectively) (Figure 3). In addition, left ventricular ejection fraction (LVEF) did not affect the diagnostic performance of CT-FFR value and CT-FFR drop using LVEF < 50% as a grouping variable (AUC CT-FFR value_LVEF < 50%_ vs. AUC CT-FFR value_LVEF ≥ 50%_: 0.736 vs. 0.813, *p* = 0.451; and AUC CT-FFR drop_LVEF < 50%_ vs. AUC CT-FFR drop_LVEF ≥ 50%_: 0.726 vs. 0.834, *p* = 0.283, respectively). Both CT-FFR value and CT-FFR drop tend to perform better in the evaluation of NCL(s) in patients with STEMI versus in patients with NSTE-ACS (*p* = 0.141 and *p* = 0.373); however, it was statistically not significant. Detailed information of vessel based diagnostic performance of anatomical and functional CCTA-parameters in subgroups is provided in Table 4 and Table 5, respectively. The overall diagnostic capacity of lesion length was close to random (0.50, CI: 0.39–0.61). The combined anatomical plus functional models (Model 1: MLDS% + CT-FFR value, AUC = 0.77, CI: 0.67–0.86; Model 2: MLDS% + CT-FFR drop, AUC = 0.77, CI: 0.67–0.85) were not better than the CT-FFR value (*p* = 0.887 and *p* = 0.815) nor the CT-FFR drop (*p* = 0.980 and *p* = 0.783) alone in the classification between ischemic and non-ischemic FFRi, respectively. From ACS to CT-FFR and CT-FFR to FFRi, 79 days (IQR: 66–100 days) and 15 days (IQR: 7–32 days) passed, respectively.

### 3.4. CT-FFR versus DSE

Accuracy, sensitivity, specificity, PPV, and NPV of CT-FFR relative to DSE was 65%, 37%, 77%, 42%, and 74%, respectively (Appendix A). Neither CCTA-based MLDS% nor CT-FFR value and CT-FFR drop showed any correlation with regional WMSI at peak stress (Appendix A). Lesion length, MLDS%, and the evaluated CT-FFR parameters were not sufficient to predict regional WMSI at peak stress even in univariate analysis (Table 3). Lesion length, MLDS%, CT-FFR value, and CT-FFR drop had poor diagnostic ability to identify NCLs with ischemia on their territory identified using DSE: AUC = 0.50, (CI: 0.40–0.61), 0.54 (CI: 0.43–0.65), 0.54 (CI: 0.43–0.64), and 0.55 (CI: 0.44–0.65), respectively (Figure 4, except lesion length). In most of the cases, CT-FFR and DSE procedures have been completed within less than two weeks relative to each other (median: 13 days, IQR: 2–24 days).

## 4. Discussion

The main findings of our study are the following: (a) on-site CT-FFR has overall moderate diagnostic performance to predict the functional significance of NCLs in ACS patients; (b) using FFRi as a reference standard, on-site CT-FFR is superior to conventional CCTA-based anatomical stenosis assessment for the functional evaluation of NCLs; (c) on-site CT-FFR cannot predict the DSE result on the territory of an NCL; (d) based on its absence of any association with DSE, on-site CT-FFR does not provide any information on the microvasculature; and (e) lesion length does not have any impact on the functional severity of an NCL.

More than 40% of patients with ACS have multivessel coronary artery disease [1]. Multiple clinical trials demonstrated that ACS patients with functionally significant residual NCL(s) after primary PCI are at higher risk for further cardiovascular events [17,18,19,20]. In addition, the long-term outcome of ACS patients with multivessel coronary artery disease but without inducible ischemia on the territory of an NCL is similar to single vessel ACS [19,34]. Based on these facts, verifying lesion-specific ischemia and determining the appropriate indication of revascularization of an NCL is crucial to improve the prognosis of this patient population. The indication or the deferral of revascularization of residual NCLs must be defined by widely accepted and recommended invasive (e.g., FFRi) or non-invasive (e.g., DSE) ischemia tests and the advantages and disadvantages of those methods must be well known; however, the role of CT-FFR in the management of moderate non-culprit coronary lesions is less studied.

### 4.1. CT-FFR versus Invasive FFR

Although the role of CT-FFR is well established in patients with stable chest pain [35], limited data is available about its usefulness in patients with ACS and coexisting moderate NCL(s) [15]. It should be noted that several different anatomical and clinical factors may affect the diagnostic capacity of CT-FFR in this patient population, which also affect its relationship with the invasively measured FFR. Invasive anatomical assessment by QCA during the acute coronary angiography can overestimate the severity of NCLs [6]. This observation is in line with our previous results [36] and we also confirmed it in the current study by a significant decrease of invasively measured MLDS% from the index episode by the time of the staged ICA. However, CCTA-based MLDS% was significantly lower compared with those measured by QCA during the staged invasive coronary angiography, in which the difference may be the result of the lower temporal and spatial resolution of CCTA in contrast to the invasive approach. Since on-site CT-FFR uses computational fluid dynamics simulations calculated from the anatomical pictures of the CCTA-scan [37], the functional underestimation of NCLs might logically be the result of the anatomical reasons mentioned above. We observed low sensitivity of on-site CT-FFR compared with FFRi and CT-FFR values were significantly higher than FFRi values, respectively. These results become even more interesting and raise questions if we consider the fact that FFRi may underestimate the functional relevance of an NCL during the acute phase and the pressure gradient over an epicardial stenosis may increase over time [6]. However, it should be noted that other authors detected no difference in FFRi values during the acute versus the staged measurements in patients with STEMI [38]. The sensitivity of on-site CT-FFR (51%) in the current study is much lower than those demonstrated in another paper (83%) with a similar aim to ours [15]. This discrepancy can be explained by several circumstances. On the one hand, the study populations were different, since Gaur et al. investigated only patients with STEMI [15], whilst we enrolled a relatively high number of patients with NSTE-ACS (33.8%), in which prior to the clinical scenario the usefulness of CT-FFR was barely investigated before [39]. On the other hand, the time between ACS and CT-scans, as well as between CT-scans and FFRi procedures, was much longer in our study than in the study published by Gaur et al. (79 days median vs. 38 days mean and 15 days median vs. 1 day mean, respectively). In our study, an unacceptably low sensitivity (40%) was observed among patients with NSTE-ACS (data are provided in the Appendix A). Although the difference in the diagnostic performance of CT-FFR between patients with STEMI and patients with NSTE-ACS was not significant in the present study, a recognizable tendency was detected in favor of its use in STEMI. These results raise the question of whether the microvascular dysfunction—which exists in both types of ACS [40,41] and can lead to the underestimation of the functional severity of an NCL—is more extensive (and maybe a chronically present pathophysiological issue) in patients with NSTE-ACS than in patients with STEMI. Further studies using the index of microcirculatory resistance during the index ACS and as a part of a staged invasive procedure on the territory of an NCL may be needed to better understand the pathophysiology of microcirculation in the territory of NCLs in different types of ACS. The timing of CT-FFR in patients with moderate NCL(s) after a recent ACS is not well established. It should be emphasized that the timing protocol in the study by Gaur et al. was based on the NXT trial (patients were excluded with ACS in their medical history within 30 days), in which only stable angina patients were enrolled. Additionally, CCTA-scan was used as a part of the initial clinical workup without delay and CT-FFR measurement was performed retrospectively in another trial by Duguay et al. with the aim to evaluate the prognostic value of CT-FFR in patients with NSTE-ACS [39]. Based on these facts, the optimal timing of CT-FFR in the assessment of NCLs, i.e., potential microvascular healing over time, can be crucial; nevertheless, it needs further validation. A moderate correlation between CCTA-based anatomical parameters and FFRi has been detected in our study, which may be interpreted from the well-known variability between anatomy and function [42]. A considerable number of papers confirmed the substantial physiological effect of lesion length on moderate coronary stenoses [43,44], which correlates well with the additive value of lesion length measured by CCTA on the identification of high-risk non-culprit plaques for a subsequent ACS [45]. Surprisingly, we did not observe this impact of CCTA-based lesion length on the functional relevance of an NCL, which even more highlights the difference between anatomy and function in this patient population and it cannot be explained only by resolution issues. Although many authors described the superiority of CT-FFR to conventional CCTA-based anatomical stenosis assessment to identify functionally relevant coronary lesions in patients with stable angina [11,12,13,46], its routine usage in patients with moderate NCL has not been supported before with unequivocal evidence [15]. Even though our study confirmed CT-FFR value and CT-FFR drop as independent predictors for FFRi, and those parameters did discriminate between ischemic and non-ischemic FFRi better than CCTA-based MLDS%, the overall diagnostic performance of on-site CT-FFR in the evaluation of angiographically borderline NCLs was moderate. This can be also interpreted by resolution issues and by the timing of the CCTA scan, respectively. Our data demonstrated better discriminative power of both lesion length and MLDS% to detect an NCL with abnormal FFRi in cases of LCx versus cases of LAD and RCA. Although the diagnostic performance of CT-FFR did not differ significantly between these subgroups, a statistically not significant tendency for the benefit of the use of CT-FFR in cases of LCx lesions was observed (LCx vs. LAD, *p* = 0.141 and LCx vs. RCA, *p* = 0.373, respectively). These results of our study, however, must be interpreted in the context of the very small sample size of investigated LCx lesions and needs further research. On-site CT-FFR is not validated in patients with heart failure. Although the major clinical trials of CT-FFR did not exclude patients with reduced LVEF, its role in the usefulness of CT-FFR is unknown [47]. In the current study, we did not find any difference in the diagnostic performance of CT-FFR parameters evaluating NCLs with reduced versus preserved LVEF. Notwithstanding that straightforward data of CT-FFR in patients with impaired LVEF are not available, we only try to explain this result with little knowledge of invasive FFR in this topic. Since right atrial pressure (as an indirect marker of heart failure) did not affect invasive FFR results [48], we presume that CT-FFR simulations are also independent from the ventricular function; however, this question should be also further investigated.

### 4.2. CT-FFR versus DSE

CT-FFR has not been prospectively validated against another non-invasive functional test; only sporadic data are available. In a retrospective analysis [49], off-site CT-FFR showed a significant discrepancy with stress echocardiography (39%); nevertheless, the number of patients who underwent both tests was quite low (*n* = 31, 19% of the whole population). In our study, we established a direct comparison of on-site CT-FFR relative to DSE, as a non-invasive reference standard with robust clinical data to identify lesion-specific ischemia in patients with moderate NCL and to determine long-term prognosis [19,20,29,31]. Our results demonstrated that on-site CT-FFR has a poor concordance with DSE (35% mismatch); it has an extremely low sensitivity (37%), PPV (42%), and poor diagnostic ability (CT-FFR value AUC = 0.54 and CT-FFR drop AUC = 0.54, respectively) to identify an ischemic DSE on the territory of an NCL. Moreover, the rule-out capacity (NPV = 74% and specificity = 77%) of on-site CT-FFR was also modest. Due to the lack of any association between CT-FFR (as well as CCTA-based anatomical markers) and DSE, we refrained from the more comprehensive comparison of them (e.g., subgroup analyses according to coronary supplying areas or the type of ACS). These results may be explained by several factors. DSE is the representative of coronary flow reserve (CFR), which is the summary of the flow in the coronary circulation, depending mainly on the microvasculature in the subendocardial layer of the myocardium [50,51]. FFRi, however, in simple terms, summarizes pressure drop through a focal epicardial coronary lesion [51]. The well-known discordance between CFR (hereby DSE) and FFRi (hereby CT-FFR) originates from these above-mentioned different basic characteristics of the two methods. In addition, both DSE and CT-FFR can be strongly influenced by numerous circumstances [e.g., local neurohumoral issues, microvascular obstruction, reduced blood flow, etc. [52,53,54]] for a long period after ACS. Furthermore, CT-FFR is independent of contractility, and it does not need maximal hyperemia, whereas the success of ischemia detection during DSE extremely depends on the increase of contractility of the myocardium and on the vasodilation in the coronary microcirculation, respectively. In summary, on-site CT-FFR and DSE are certainly on different stages of evaluating the ischemic cascade of the myocardium. Nevertheless, our results should not be considered as a validation process of on-site CT-FFR as opposed to the non-invasive DSE.

### 4.3. Limitations

Several limitations of our study must be mentioned. Firstly, it is a single-center study. Secondly, non-hyperemic pressure gradients (e.g., instantaneous wave-free ratio or resting full cycle ratio) data are not available; however, it can be an interesting area of research to compare on-site CT-FFR to another functional tool that does not require maximal hyperemia. Thirdly, contrast agent was used only in patients with poor image quality during DSE, which could have improved image interpretation and the sensitivity of the test [55]. Fourthly, power calculation was not performed before the study. Finally, we lost a relatively high number of patients during the screening period or they failed to attend CT-scans, which may cause a selection bias. The presence of selection bias limits the generalizability of our findings; therefore, further studies are warranted to explore the value of CT-FFR in patients with NCL(s) after a recent ACS.

## 5. Conclusions

In summary, on-site CT-FFR is superior to conventional CCTA in the functional assessment of moderate non-culprit coronary lesions in patients with acute coronary syndrome; however, its overall diagnostic performance, compared to invasive fractional flow reserve, is limited. Based on these observations, on-site CT-FFR, at this stage of development, might be insufficient to be used as a gatekeeper to the invasive functional evaluation in this clinical setting. Moreover, as on-site CT-FFR has no association with DSE whatsoever, it probably does not provide information on the microvascular dysfunction in the territory of non-culprit coronary lesions.

## Figures and Tables

**Figure 1 life-12-01820-f001:**
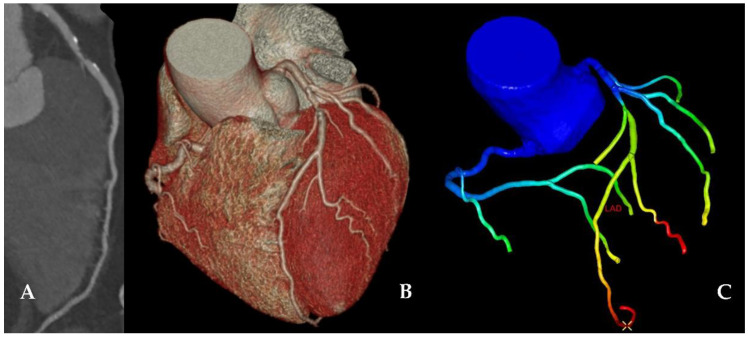
From conventional CCTA to on-site CT-FFR simulation: a representative sample. Panel (**A**): conventional CCTA of an LAD; Panel (**B**): 3D modelling of the major coronary arteries; Panel (**C**): CT-FFR simulation (the warmer the color, the lower the CT-FFR value). CT-FFR value can be determined in each point of the epicardial coronary tree. Abbreviations: CCTA—coronary computed tomography angiography; CT-FFR—computed tomography derived fractional flow reserve; LAD—left anterior descending artery.

**Figure 2 life-12-01820-f002:**
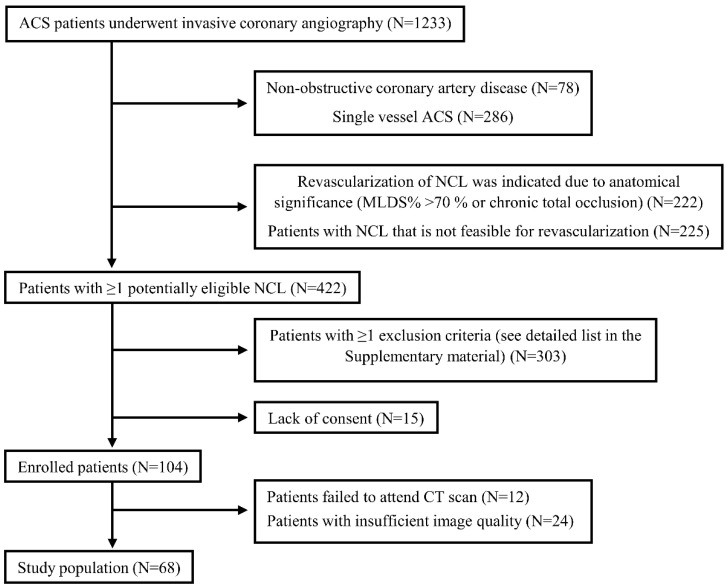
Patient selection process. N represents the number of patients.

**Figure 3 life-12-01820-f003:**
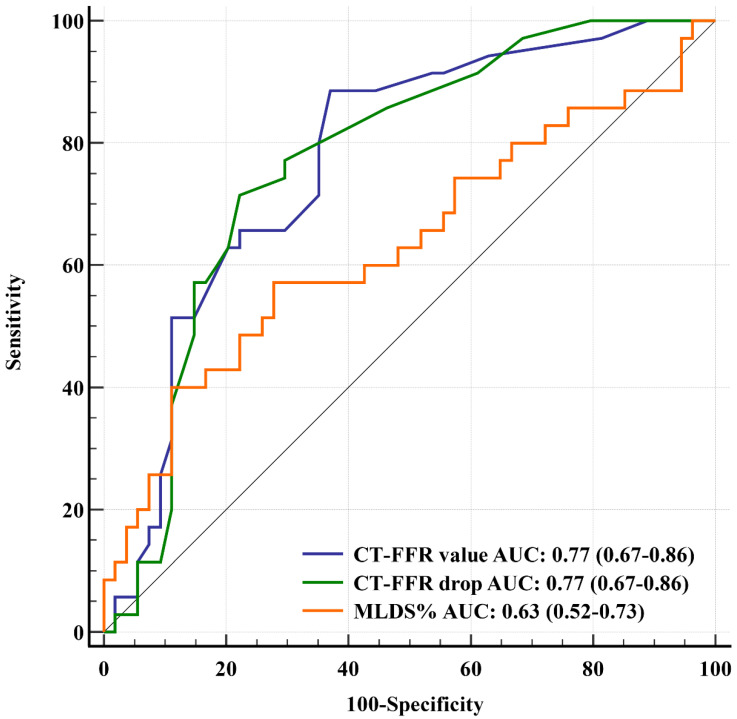
ROC curves of anatomical and functional CCTA-based parameters using FFRi as reference standard.

**Figure 4 life-12-01820-f004:**
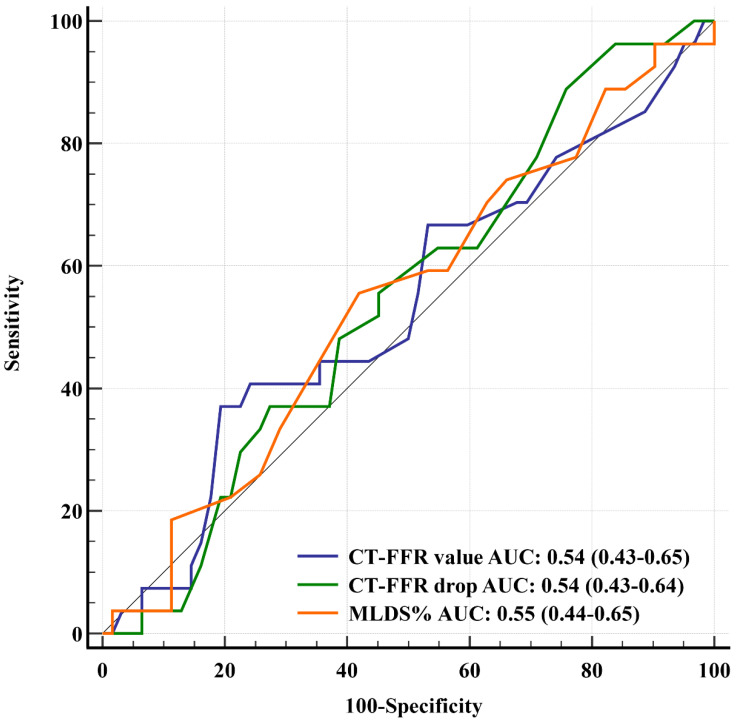
ROC curves of anatomical and functional CCTA-based parameters using DSE as reference standard.

**Table 1 life-12-01820-t001:** Baseline characteristics of the study population.

	*n* = 68
Demography and cardiovascular risk factors	
Female (*n*, %)	22 (32.4)
Age (years)	57.1 ± 10.1
BMI (kg/m^2^)	28.1 ± 4.1
Hypertension (*n*, %)	44 (64.7)
Diabetes mellitus (*n*, %)	16 (23.5)
Hyperlipidaemia (*n*, %)	36 (52.9)
Current smoker (*n*, %)	40 (58.8)
Family history of cardiovascular disease (*n*, %)	34 (50.0)
Medication	
ASA (*n*, %)	67 (98.5)
P2Y12 ADP receptor blocker (*n*, %)	68 (100)
Beta-blocker (*n*, %)	67 (98.5)
Statin (*n*, %)	67 (98.5)
LVEF (%)	47.3 ± 8.4
LVEF < 50% (*n*, %)	32 (47.1)

Abbreviations: ADP—adenosin-disphosphate; ASA—acetylsalicylate; BMI—body mass index; LVEF—left ventricular ejection fraction.

**Table 2 life-12-01820-t002:** Major CCTA-based anatomical and functional findings of the investigated non-culprit lesions.

	Overall	LAD	LCx	RCA	*p*
	*n* = 89	*n* = 52	*n* = 15	*n* = 22	
Lesion length (mm)	22.5 ± 13.4	24.0 ± 14.2	19.1 ± 9.0	21.4 ± 13.9	0.511
MLDS%	39.4 ± 16.2	39.5 ± 16.0	42.0 ± 20.4	37.4 ± 14.0	0.707
MLDS% ≥ 50% (*n*, %)	20 (22.5)	12 (23.1)	4 (26.7)	4 (18.2)	0.821
CT-FFR value	0.85 ± 0.09	0.86 ± 0.09	0.87 ± 0.09	0.84 ± 0.10	0.479
CT-FFR drop	0.12 ± 0.09	0.12 ± 0.09	0.11 ± 0.09	0.11 ± 0.10	0.763
CT-FFR value ≤ 0.80 (*n*, %)	24 (27.0)	14 (26.9)	4 (26.7)	6 (27.3)	0.999

*p* value represents the result of comparisons between the investigated subgroups of major epicardial coronary arteries. Abbreviations: CCTA—coronary computed tomography angiography; CT-FFR—Computed tomography derived fractional flow reserve; CT-FFR drop—the difference of CT-FFR values between proximal and distal measuring points (proximal minus distal); LAD—left anterior descending coronary artery; LCx—left circumflex coronary artery; MLDS%—degree of minimal lumen diameter stenosis in percentages, RCA—right coronary artery.

**Table 3 life-12-01820-t003:** Univariate and multivariate linear regression analysis of conventional CCTA and CT-FFR parameters. FFRi value and regional WMSI at peak stress were used as standard markers. The multivariate regression Model 1 and Model 2 separately contain the CT-FFR value and the CT-FFR drop to avoid multicollinearity.

	FFRi Value						Regional WMSI at Peak Stress	
	Univariate		Multivariate Model 1		Multivariate Model 2		Univariate	
	Unstandardized Beta	*p*	Unstandardized Beta	*p*	Unstandardized Beta	*p*	Unstandardized Beta	*p*
Lesion length	−0.001	0.177	-	-	-	-	0.000	0.956
MLDS%	−0.001	0.007	0.000	0.454	−0.001	0.233	−0.004	0.221
CT-FFR value	0.366	<0.001	0.334	<0.001			−0.440	0.441
CT-FFR drop	−0.337	<0.001	-	-	−0.289	0.002	0.403	0.494

Abbreviations: CCTA—coronary computed tomography angiography; CT-FFR—computed tomography derived fractional flow reserve; CT-FFR drop—the difference of CT-FFR values between proximal and distal measuring points (proximal minus distal); FFRi—invasive fractional flow reserve; MLDS%—degree of minimal lumen diameter stenosis in percentages.

**Table 4 life-12-01820-t004:** Diagnostic performance defined by AUC values of anatomical and functional CCTA-based parameters in subgroups divided by major coronary vessels.

	LAD	LCx	RCA
	*n* = 52	*n* = 15	*n* = 22
Lesion length	0.54 (0.39–0.68) *	0.89 (0.62–0.99) *	0.59 (0.36–0.79)
MLDS%	0.57 (0.42–0.71) #	0.89 (0.62–0.99) #	0.66 (0.43–0.84)
CT-FFR value	0.75 (0.61–0.86)	0.91 (0.65–1.0)	0.79 (0.57–0.93)
CT-FFR drop	0.77 (0.63–0.88)	0.91 (0.65–1.0)	0.70 (0.47–0.87)

* *p* = 0.005; # *p* = 0.010; Other comparisons between the investigated groups were statistically not significant at the *p* < 0.05 level. Abbreviations: AUC—area under the curve; CCTA—computed coronary tomography angiography; CT-FFR—computed tomography derived fractional flow reserve; CT-FFR drop—the difference of CT-FFR values between proximal and distal measuring points (proximal minus distal); FFRi—invasive fractional flow reserve; LAD—left anterior descending coronary artery; LCx—left circumflex coronary artery; MLDS%—degree of minimal lumen diameter stenosis in percentages; RCA—right coronary artery.

**Table 5 life-12-01820-t005:** Vessel-based diagnostic performance (AUC values) of CCTA-based anatomical assessment tools and CT-FFR parameters in different types of ACS in the detection of a functionally relevant NCL using FFRi as reference standard.

	STEMI	NSTE-ACS	*p*
	*n* = 57	*n* = 32	
Lesion length	0.58 (0.44–0.71)	0.58 (0.40–0.76)	0.951
MLDS%	0.64 (0.51–0.77)	0.62 (0.43–0.78)	0.837
CT-FFR value	0.82 (0.70–0.91)	0.65 (0.46–0.81)	0.141
CT-FFR drop	0.81 (0.68–0.90)	0.70 (0.52–0.85)	0.373

Abbreviations: ACS—acute coronary syndrome; AUC—area under the curve; CCTA—computed coronary tomography angiography; CT-FFR—computed tomography derived fractional flow reserve; CT-FFR drop—the difference of CT-FFR values between proximal and distal measuring points (proximal minus distal); FFRi—invasive fractional flow reserve; MLDS%—degree of minimal lumen diameter stenosis in percentages; NSTE-ACS—non-ST-elevation acute coronary syndrome; STEMI—ST-elevation myocardial infraction.

## Data Availability

The data that support the findings of this study are available from the corresponding author (P.A.) upon reasonable request due to privacy restrictions.

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
