# Peer review of "Diagnostic Performance of On-Site Computed Tomography Derived Fractional Flow Reserve on Non-Culprit Coronary Lesions in Patients with Acute Coronary Syndrome"

_life, 2022, doi:10.3390/life12111820_

Round 1

Reviewer 1 Report

This study evaluates the diagnostic performance of on-site computed tomography-derived fractional flow reserve on non-culprit coronary lesions in patients with acute coronary syndrome.

I have some comments for authors to consider

1. The introduction needs to be expanded with some background on the diagnostic performance of CCTA and other modalities based on previous studies. 

2. Lines 40-42: Authors say, "...present with angiographically moderate non-culprit lesions" - please clarify what you mean by "moderate"?

3. Methods: Please clarify if this is a prospective or retrospective study.

4. Statistical analysis: It is not clear if sample size calculation was performed for power estimation?

5. 24 patients were excluded due to poor image quality. Given the small number of the study size, this is sizable. Could the authors clarify how was th quality of image assessed? How many investigators were responsible for the image quality assessment?

6. One of the major shortcomings of this study is the selection bias - and this should be discussed in limitations. In light of this, the authors should clarify that the findings of this study should be interpreted in the context of the study design, and needs further validation.

Furthermore, this study is not optimal for clinical practice recommendations - so any observation made should be interpreted in this context, requiring further investigation.

Author Response

Uploaded as a Word file.

Reviewer 2 Report

In this prospective clinical study, the authors evaluated the diagnostic performance of on-site computed tomography (CT) derived fractional flow reserve (FFR) on non-culprit coronary lesions (NCL) in patients with acute coronary syndrome (ACS). The main findings of the authors are that on-site CT-FFR has overall moderate diagnostic performance to predict the functional significance of NCLs in ACS patients and that, using FFRi as reference standard, on-site CT-FFR is superior to conventional CCTA-based anatomical stenosis assessment for the functional evaluation of NCLs. However, in this study, on- site CT-FFR cannot predict the dobutamine stress echocardiography (DSE) result on the territory of an NCL. This study addresses a current topic regarding the role of non-invasive techniques in the assessment of hemodynamic relevance of NCLs in patients with previous ACS.

1.       In Table 1, baseline characteristics of the study population are presented. The authors should include data describing ventricular function. Does ventricular function affect the performance of on-site CT-FFR?

2.       In Table 2, the authors should also include the CCTA-based anatomical and functional findings of the investigated NCLs by distinguishing between LAD, LCx and RCA. The authors should address the question, if the performance of on-site CT-FFR may be influenced by the affected coronary vessel.

3.       In the Discussion, the authors state that “The sensitivity of on-site CT-FFR (40%) in the current study is much lower than those demonstrated in another paper (83%) with a similar aim to ours [13]. On the one hand, this discrepancy can be explained by the different study population, since Gaur et al. investigated only patients with STEMI [13], whilst we enrolled a relatively high number of patients with NSTE-ACS (33.8%)”.

Previous studies already showed that, in comparison to patients with stable angina (Koo et al., . J Am Coll Cardiol 2011; Min et al., JAMA 2012; Nørgaard  et al., J Am Coll Cardiol 2014), the diagnostic performance of CT-FFR in patients post-STEMI is lower (Gaur et al. JACC: Cardiovascular Imaging 2017). Could the authors find any relevant differences in the diagnostic performance of CT-FFR in STEMI-patients compared to NSTEMI patients? How can the authors explain these findings? A detailed analysis of this relevant point should be included in the discussion.

4.       In the Discussion, the authors state that “the timing of CT-FFR in the assessment of NCLs can be crucial”. Could the authors find any relevant differences in the diagnostic performance of CT-FFR in this study which are related to the timing of CT-FFR? The authors should better elucidate the role of timing of CT-FFR in the evaluation of hemodynamic relevance of NCLs in the discussion.

Author Response

Provided as a Word filw

Reviewer 3 Report

In the present study Dr Ahres and coworkers presented the diagnostic accuracy of on-site CT-FFR measurement in non culprit lesions of patients suffering an ACS at least 30 days earlier, compared with invasive FFR as gold standard. The diagnostic accuracy of CT-FFR was moderate (70%)

Comments: 

1. The authors provide an excellent presentation of the methods used for modality

2. I would suggest an explanatory figure for the on-site CT-FFR measurement. This would be helpful for a novel reader

3. Has this prototype of on-site CT-FFR measurement been validated in previous studies? How long do the calculations take to provide a result? Dows this software provide CT-FFR values for all coronary segments (like Heartflow?)

4. In figure 1 it is mentioned that 24 pts were not eligible for the study due to poor image quality. This number appears high (almost 25% of the pts) for a latest generation CT scanner. Please comment

5. The findings that only 22.5% of the stenosis were >50% but almost 40% showed FFRi values <0.8 (compatible with ischemia), together with the low sensitivity of CT-FFR are very interesting. The time between ACS and CT was >1 month, how could these findings be explained?

6. Why did the authors choose 30% cutoff for luminal stenosis and not 40% as suggested by the guidelines?

Author Response

Uploaded as a Word file

Round 2

Reviewer 1 Report

No further comments

Reviewer 2 Report

All my points have been addressed, I have no further comments.

Reviewer 3 Report

Comments adequately addressed